# DISENTANGLING TIME SERIES REPRESENTATIONS VIA CONTRASTIVE INDEPENDENCE-OF-SUPPORT ON $l$-VARIATIONAL INFERENCE

**Khalid Oublal**[*†‡]**, Saïd Ladjal**[†]**, David Benhaiem**[‡]**, Emmanuel Le-borgne**[‡]**, François Roueff**[†]

[†]Institute Polytechnique de Paris, Telecom Paris LTCI/S2A, [‡]OneTech TotalEnergies, DS&AI

[*]Primary contact: khalid.oublal@polytechnique.edu

## ABSTRACT

Learning disentangled representations for time series is a promising path to facilitate reliable generalization to in- and out-of distribution (OOD), offering benefits like feature derivation and improved interpretability and fairness, thereby enhancing downstream tasks. We focus on disentangled representation learning for home appliance electricity usage, enabling users to understand and optimize their consumption for a reduced carbon footprint. Our approach frames the problem as disentangling each attribute's role in total consumption. Unlike existing methods assuming attribute independence which leads to non-identifiability, we acknowledge real-world time series attribute correlations, learned up to a smooth bijection using contrastive learning and a single autoencoder. To address this, we propose a **D**isentanglement under **I**ndependence-**O**f-**S**upport via **C**ontrastive Learning (`DIOSC`), facilitating representation generalization across diverse correlated scenarios. Our method utilizes innovative $l$-variational inference layers with self-attention, effectively addressing temporal dependencies across bottom-up and top-down networks. We find that `DIOSC` can enhance the task of representation of time series electricity consumption. We introduce `TDS` *(Time Disentangling Score)* to gauge disentanglement quality. TDS reliably reflects disentanglement performance, making it a valuable metric for evaluating time series representations disentanglement. Code available at https://institut-polytechnique-de-paris.github.io/time-disentanglement-lib.

## 1 INTRODUCTION

Disentangled representation learning is crucial in various fields like computer vision, speech processing, and natural language processing (Bengio et al., 2014). There have been efforts to learn disentangled time series representation (Woo et al., 2022; Yao et al., 2022), with the aim to improve generalization, robustness, and explainability. A core task in representation learning is provable representation identification. We call a representation disentangled when identified attributes in the data are specifically coded in the structure of its latent units. How this can be achieved remains an open research question. In (Locatello et al., 2019), it is shown

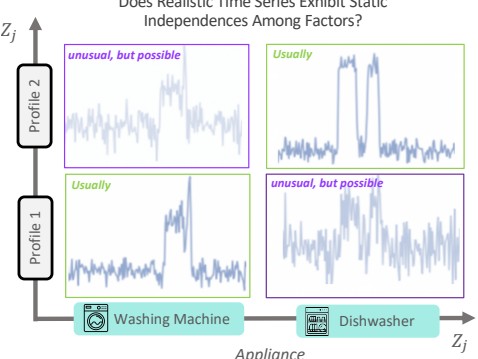

Figure 1: Time series real-world often showcases attributes exhibiting strong correlation.

that disentangling requires some kind of supervised learning and inductive bias. Moreover, standard methods such as $\beta-$`VAE` (Higgins et al., 2016), `TCVAE` (Chen et al., 2018c), and rely on the too stringent assumption of statistical independence among ground truth attributes. In real-world time series, attributes are often correlated. In the application of this study, the attributes correspond to the contributions of specific devices in an aggregated consumption signal. We illustrate the correlation of the attributes in Fig 1, where green boxes contain typical consumption for their respective appliances

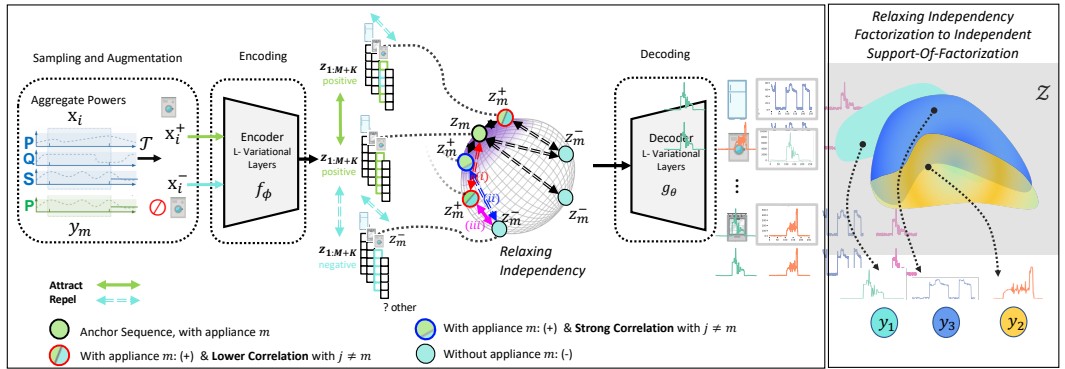

Figure 2: **Disentanglement under Independence-of-Support via Contrastive (`DIOSC`)**, Representations of positive pairs attract each other, while negative repels their corresponding representations. The latent attributes exhibit causal correlations (Shanmugam, 2018), `DIOSC` allows for scenarios where unlikely (but exist in data) combinations occur ((i) and (ii) leading to the existence of (iii)). It's worth noting that forcing strict statistical independence does not prevent these cases.

and purple ones show consumption profiles that are correlated to those of the other appliances. Existing methods in this context, such as $\beta$-`VAE`, `TCVAE`, often assume independent attributes. This paper provides a unified framework for disentangling time series by relaxing the assumption of statistical independence in the latent representation. To illustrate this, we will focus on a crucial application of time series disentanglement: household energy consumption disaggregation, also known as Non-Intrusive Load Monitoring (NILM). Given only the main consumption of a household $\mathbf{x} \in \mathbb{R}^{c \times T}$, seen as a $c$-variate time series observed at times $t = 1, \ldots, T$, the NILM algorithm identifies the active consumption $y_m \in \mathbb{R}^T$ of each operating appliance $m$. Such a task has received a growing interest and still raises unsolved problems. The fact that many households rely on past bills to adjust future energy use underscores the importance of energy disaggregation algorithms in reducing carbon footprints. Recent work (Bucci et al., 2021; Nalmpantis & Vrakas, 2020) hold promising results, yet challenges in generalization to in- and out-of distribution. In this paper, we address NILM with a disentanglement perspective, we assume that different appliance $m$ need to have latent $z_m \in \mathbb{R}^d$. On downstream disaggregation task, we show that a disentangled improves generalization to distribution shifts. We draw on connection between contrastive learning and identifiability in the form of Nonlinear ICA (Jutten et al., 2010; Hyvärinen, 2013).

**Contributions and Main results.** Our approach stands out by relying on weak contrastive learning using support factorization on the prior (rather than strict statistical independence) and Attentive $l$-Variational Inference. We evaluate our method qualitatively and quantitatively across various datasets with ground-truth labels, examining the generalization capabilities of the learned representations on correlated data. In summary:

[1] We define a new regularization term and its theoretical justification, `DIOSC`, whose goal is to address latent space misalignment issues and to preserve disentanglement. This is achieved by promoting specific *Pairwise (dis-)similarities* over the latent sub-variables (c.f. § 4).

[2] Our experiments across three datasets and diverse correlation scenarios demonstrate that DIOSC significantly enhances robustness to attribute correlation, yielding up to a **+61.4%** for reconstruction and **+21.7%** improvement in disentanglement metrics (`RMIG`, `DCI`, and `TDS`) compared to state-of-the-art methods (c.f. § 5.3).

[3] We propose $l$-variational-based self-attention for extracting high semantic representations from time series, ensuring complex representations without temporal locality.

[4] To evaluate disentanglement we proposed `TDS` score, aling with the performance in the downstream task. We implemented our framework in a user-friendly library, making it the first time-series disentanglement framework.

## 2    RELATED WORK

**Time Series Disentanglement in the Realm of Correlated Attributes.** Traditional methods for time series disentanglement often emphasize enforcing statistical independence among representation dimensions (Do & Tran, 2021; Klindt et al., 2020), even when dealing with highly correlated data. In recent computer vision disentanglement methods, there has been an exploration of using auxiliary information to improve identifiability, moving away from the assumption of statistical

independence (Roth et al., 2023). However, both (Träuble et al., 2021; Roth et al., 2023) point out the limitations of this approach due to non-identifiability. Another study by (Wang & Jordan, 2022a) proposes support factorization for disentanglement from a causal perspective, incorporating a Hausdorff objective akin to (Roth et al., 2023). In our unique approach, we tackle time series disentanglement without explicit auxiliary variables or prior models. Instead, we achieve pairwise factorized support through contrastive learning, departing from the traditional independence assumption. Recent contributions (Wang & Jordan, 2022b; Roth et al., 2023) seek to alleviate this assumption, yet remain disconnected from observational data and grapple with numerical stability. This method pioneers disentanglement in correlated time series by emphasizing independence-of-support through contrastive learning during training. This contrasts with methods like (Ren et al., 2021), where representation discovery relies on contrastive learning of pre-trained generative models with assumed independence factorization during training. To the best of our knowledge, the present work provides the first identifiability and disentanglement result for time series in real correlated scenarios.

**On The Non-Intrusive Load Monitoring and Representation Learning.** Recent work (Bucci et al., 2021; Nalmpantis & Vrakas, 2020) has produced promising results for separation source power. Nevertheless, they encounter challenges related to generalization and robustness when confronted with out-of-distribution scenarios. Several approaches have been suggested to address these challenges. Some methods tackle them through either transfer learning or by enhancing the learned representations for each individual appliance. Exploring ways to enhance representation learning in this field has been the focus of recent studies (Woo et al., 2022; Vahdat & Kautz, 2021; Maaløe et al., 2019). However, achieving an informative and disentangled representation remains an open and challenging question. Existing models, like `RNN-VAE` (Chung et al., 2015) for sequential data and `D3VAE` (Li et al., 2023), assume statistically independent attributes. This assumption hampers their performance on real-world data and makes them less applicable to out-of-distribution scenarios. Developing models that effectively capture informative and disentangled representations in a realistic and versatile manner continues to be a significant challenge.

## 3 PROBLEM STATEMENT AND PRELIMINARIES

Our approach belongs to the general framework of Variational Auto-Encoders VAEs, and thus relies on two main ingredients: 1) a variational family $(q_\phi)$, which approximates the conditional density of the latent variable given the observed variable based on an encoder $f_\phi$; 2) a generative model $(p_\theta)$ based on a latent variable, and a decoder $g_\theta$. We consider a $C$-variate time series observed at times $t = 1, \ldots, T$, we denote by $\mathbf{x} \in \mathbb{R}^{C \times T}$ the $C \times T$ resulting matrix with rows denoted by $x_1, \ldots, x_C$. Each row can be seen as a univariate time series. The goal is to recover the following decomposition of the active power $x_{c=1} = \mathbf{y} + \xi$ , where $\mathbf{y}$ is a matrix with $M$ columns $y_m \in \mathbb{R}^T$ denotes the contribution of the $m$-th electric device, among the total of $M$ devices identified, and $\xi \in \mathbb{R}^T$ contains the contribution of $K$ unknown sources and/or additive noise. The NILM mapping, denoted as $\mathbf{x} \mapsto \mathbf{y}$, is typically learned from a training set $\mathcal{X} = \{\tilde{\mathbf{x}}_i\}_{i=1}^M$, where each $\tilde{\mathbf{x}}_i = (\mathbf{x}_i, \mathbf{y}_i)$ represents a pair of input-output samples used for training purposes. In a VAE, both (unknown) parameters $\theta$ and $\phi$ are learnt from the training set $\mathcal{X}$. A key idea for defining the goodness of fit part of the learning criterion is to rely the **E**vidence **L**ower **B**ound (ELBO), which provides a lower bound on (and a proxy of) the log-likelihood

$$\log p_\theta(\tilde{\mathbf{x}}) \geq \mathbb{E}_{q_\phi(\mathbf{z}|\tilde{\mathbf{x}})} \left[\log p_\theta(\tilde{\mathbf{x}}|\mathbf{z})\right] - \text{KL}(q_\phi(\mathbf{z}|\tilde{\mathbf{x}}) \parallel p(\mathbf{z})) \, , \quad (3.1)$$

where we denoted the latent variable by $\mathbf{z}$, defined as a $d_z \times (M+K)$ matrix and $p$ denotes its distribution. The use of ELBO goes back to traditional variational Bayes inference. The encoder $f_\phi$ provides an approximation of $\mathbf{z} = \{z_1, \ldots, z_{M+K}\} \sim p_\mathbf{z}$ from $\mathbf{x}$ while $\mathbf{y} := g_\theta(\mathbf{z})$. A standard choice in a VAE is to rely on Gaussian distributions and, for instance, to set $q_\phi(\mathbf{z}|\mathbf{x}) = \mathcal{N}(\mathbf{z}; \mu(\mathbf{x}, \phi), \sigma^2(\mathbf{x}, \phi))$, where $\mu(\mathbf{x}, \phi)$ and $\sigma^2(\mathbf{x}, \phi)$ are the outputs of the encoder $f_\phi$. As discussed in Section 1, various criterion functions such as $\beta$/TC/Factor/DIP-VAE have been introduced, aiming to learn a disentangled latent variable $\mathbf{z}$ and align it with the corresponding attributes. However, these methods typically assume statistical independence among attributes, leading to the assumption: $p(\mathbf{z}) = p(z_1) \ldots p(z_{M+K})$. In the real world, this assumption does not hold, appliances are not used independently; rather, they are used simultaneously, and their profiles may exhibit correlation (though less likely), thereby challenging the validity of Independent Factorization.

**Definition 1. Independence-of-Support Factorization (IOS).** For a latent variable $\mathbf{z} = \{z_1, \ldots, z_{M+K}\}$ sampled from $p(\mathbf{z})$, if $\mathcal{Z} = \mathcal{Z}_1 \times \ldots \times \mathcal{Z}_{M+K}$, where $\mathcal{Z}$ is the support of $p(\mathbf{z})$, and $\mathcal{Z}_j$ denotes the supports of marginal distributions of $z_j$, then $\mathbf{z}$ exhibits Independence-of-Support.

To address this, we propose a contrastive pairwise similarity to strengthen the constraint for better representation identification with observational data. As a starting point, we assume that observed data samples $\mathbf{y}$ of appliance powers are generated from a set of latent random vectors $\mathbf{z}$ through a diffeomorphism[1] $g_\theta : \mathcal{Z} \rightarrow \mathcal{X}$, mapping from a *latent* space $\mathcal{Z}$ to an *observation* space $\mathcal{Y} \subset \mathcal{X}$,

$$\mathbf{z} \sim p_{\mathbf{z}}, \qquad \mathbf{y} = g_\theta(\mathbf{z}). \tag{3.2}$$

The only assumption we place on $p_{\mathbf{z}}$ is that it is fully supported on $\mathcal{Z}$. In particular, we do not require independence and allow for arbitrary dependencies between components of , motivated by the fact that the properties of certain time series profiles may be correlated with those of other profiles.

## 4 DISENTANGLING UNDER IOS VIA CONTRASTIVE THEORY

Under the problem of non-identifiability, to solve the NILM problem effectively in real-world scenarios, we seek an ideal disentangled representation. We use a single encoder/decoder for simultaneous and more efficient disentangling. To achieve this, we employ the following strategies: 1) Disentanglement and Independence-of-Support via Contrastive Learning (`DIOSC`), which first promotes similarity between the latent representations $z_m$ when the device $m$ is present in both $\mathbf{x}_i$ and its augmentation while inducing dissimilarity in negative cases; 2) we propose using an Attentive *l*-Variational Auto-Encoder integrate self-attention mechanisms (Vaswani et al., 2017), to enhance the model's ability to capture intricate patterns and achieve robust reconstruction. In the upcoming sections, we will delve into the theoretical underpinnings of our proposed methods.

### 4.1 DISENTANGLING UNDER INDEPENDENT-OF-SUPPORT VIA CONTRASTIVE (DIOSC)

Given appliance power sampled from the generative model outlined in Eq. (3.2), we now seek to understand under what conditions an *inference model* $\hat{f}_\phi : \mathcal{X} \rightarrow \mathcal{Z}$ will provably identify the ground-truth latent representations. Ideally, we would like $\hat{f}_\phi$ to recover the true inverse $f_\phi := g_\theta^{-1}$, but that is generally only possible up to certain irresolvable ambiguities. In our NILM setting, the objective is to separate the power representations such that each inferred latent captures *one and only one* ground-truth latent as one individual latent block, we can learn a fully disentangled representation. To this end, we define block affine identification: the true latent variables $\mathbf{z}$ are *block-identified* by a function $f_\phi : \mathcal{X} \rightarrow \mathcal{Z}$ if the inferred latent $\hat{\mathbf{z}} = f_\phi(\mathbf{x})$ contains *all and only* information about $\mathbf{z}$, i.e., if there exists some smooth *invertible* mapping $\Gamma : dim(\mathcal{Z}) \rightarrow dim(\mathcal{Z})$ s.t. $\mathbf{z} = \Gamma(\hat{\mathbf{z}})$. This identification can be connected to disentnaglement under IOS Def. 1.

**Theorem 1 (Identifiability and Disentanglement).** *Suppose the observational data is generated from Eq. (3.2) under the following assumptions:*

(i) *The support of $p_{\mathbf{z}}$ satisfies Def. 1, and interior of the support is a non-empty subset of $\mathbb{R}^d$, and for $n \neq m$, a pair $(\hat{z}_n, \hat{z}_m)$ satisfies $\hat{\mathcal{Z}}_{n,m} = \hat{\mathcal{Z}}_n \times \hat{\mathcal{Z}}_m$.*

(ii) *$\mathbf{x}$ satisfies a positivity condition: for every $m$, we have $p(z_m|\mathbf{x}) > 0$ if and only if $p(z_m) > 0$; under this condition, if the representation $\mathbf{z} = (z_1, \ldots, z_{M+K})$ is disentangled, then: The support of each dimension $m$ remains unchanged whether conditioned on other dimensions $n \neq m$ or not.*

*A unique disentangled representation is then defined by the autoencoder $(f_\phi, g_\theta)$ that solves Eq. (4.4) achieves **Permutation, Translation**, and **Scaling** identification, i.e., $\forall z \in \mathcal{Z}, \hat{\mathbf{z}} = \Lambda \Pi \mathbf{z} + Const.$, where $\hat{\mathbf{z}}$ is the output of the encoder $f_\phi$, $\mathbf{z}$ is the true latent and $\Pi$ is a permutation matrix and $\Lambda$ is an invertible diagonal matrix.*

To demonstrate the validity of Thm. 1, we rely on (Thm. 5.3, Ahuja et al. (2023)) and (Thm. 9, Wang & Jordan (2022b)). The full proof is provided in App. A. Thm. 1 demonstrates that ensuring independence between the supports of latent variables is key to achieving identification in observational data, allowing for permutation, shift, and scaling transformations. Representation encodes the same data properties, specifically, two representations $\mathbf{z}$ and its augmented version $\mathbf{z}^+$

---

[1]A bijective function between differentiable manifolds that is smooth and the inverse is also smooth.

satisfy the same sigma algebra $\sigma(\mathbf{z}) = \sigma(\mathbf{z}^+)$ if a bijective function $\Gamma$ exists such that $\mathbf{z} = \Gamma(\mathbf{z}^+)$. To achieve this, we use a contrastive objective to learn representations that enforce identifiability and disentanglement via support independence, as leaverged in Thm. 1. Additionally, this perspective offers an alternative understanding of identifiability of Zimmermann et al. (2022), as it has been shown for the hypersphere, and convex bodies $\mathcal{Z}$, the minimization of the adapted objective function $\mathcal{L}_{CL}$ solve the unmixing problem of non-linear ICA.

$$\mathcal{L}_{CL}(f; \tau, N) := \mathbb{E}_{(\mathbf{x}, \mathbf{x}^+) \sim p_{\text{positive}} \{\mathbf{x}_i^-\}_{i=1}^N \overset{\text{i.i.d.}}{\sim} p_{\text{data}}} - \log \frac{e^{f_\phi(\mathbf{x})^\top f_\phi(\mathbf{x}^+)/\tau}}{e^{f_\phi(\mathbf{x})^\top f_\phi(\mathbf{x}^+)/\tau} + \sum_{i=1}^N e^{f_\phi(\mathbf{x})^\top f_\phi(\mathbf{x}_i^-)/\tau}}. \quad (4.1)$$

Here $N \in \mathbb{Z}_+$ is a fixed number of negative samples, $p_{\text{data}}$ is the distribution of all observations and $p_{\text{positive}}$ is the distribution of positive pairs. In this equation the sum in the numerator extends to all $N$ negative pairs generated and its size depends on the batch size. In multiclass settings like NILM, the contrastive loss (Eq. 4.1) struggles when multiple samples share the same appliance latent $z_m$ due to labeled data. A suggested generalization (Khosla et al., 2021) tackles labeled cases. However, drawbacks include the lack of $\mathbf{z}$ invariance and challenges with limited or noisy labels, especially in obtaining both negative and positive labels for time series.

**Disentanglement, Invariant, and Axis-Alignment Latents.** When the $m$-th component of $\mathbf{z}$, denoted by $z_m$, remains invariant regardless changes in $\mathbf{x}$, $z_m$ is meaningless and contains no information from $\mathbf{x}$. We consider that the latent space *aligned* when the variations of latent variables $z_m$ only have an influence on the $m$-th output of the decoder $g_\theta$ applied to $\mathbf{z}$. Both terms form the basis of the disentanglement principle. Thus, $\mathbf{z}$ is considered disentangled when there is a one-to-one correspondence between each ground truth $y_m$ and the corresponding $z_m$ in the representation. To uphold this despite the constraint of limited labels, hard to get both negative and positive in NILM, and without relying on static attribute independence, we leverage weak contrastive learning (Zimmermann et al., 2022; Zbontar et al., 2021), and adjusting it for disentanglement. Rather, to overcome the constraints of Eq. (4.1) and establish invariance and alignment in a single step, we extend a contrastive objective of Zbontar et al. (2021). Our objective, integrates two core components: *latent-Invariant* component seeks to minimize information overlap between $z_m$ and its negatives $z_m^-$; *latent-Alignment* compelling similarity between $z_m$ and its augmented $z_m^+$ accommodating potential changes to cover the variability factor of variation in ground-truth attributes, ensuring both *invariance*, *alignment* and enabling the discrimination task. To further enforce (i)-Thm. 1) empirically, we demonstrate how this constraint could encompass an extra assumption relaxing IOS Asm. 4.1.

**Assumption 4.1 (Empirical Relaxing Independent Factorization to IOS).** Consider an empirical support $\mathcal{Z} \approx \mathbf{Z}$ where $\mathbf{Z} = \{\mathbf{z}_i\}_{i=1}^b$, and $b$ mini-batch size. Forcing IOS implies that $\mathbf{Z}$ aligns with its Cartesian product $\mathbf{Z}^\times$. Therefore, we minimize sliced/pairwise contrastive approximating $\mathbf{Z}$ and $\mathbf{Z}^\times = z_{:,1} \times \ldots \times z_{:,M+K}$, where $z_{:,m} \in \mathbb{R}^{b \times d}$.

Building upon Asm. 4.1, which incorporates the *invariant* and *alignment* properties, we impose a constraint on a given mini-batch $\mathbf{Z}$ of size $b$. Specifically, we ensure that the elements $\mathbf{Z}_{:,m}$ are close to their corresponding augmented $\mathbf{Z}_{:,m}^+$ and far from any negative $\mathbf{Z}_{:,m}^-$, while simultaneously preserving the independence of the Support (IoS). This involves minimizing the distance between sets $\mathbf{Z}_{:(m,\neq m)}$ and $\mathbf{Z}_{:,m} \times \mathbf{Z}_{:,\neq m}$ for all appliances $m$. Owing the discriminating nature of contrastive learning over data, this IoS constraint can be met by focusing on contrastive learning $\mathbf{Z}_{:,m}$ and its augmentation $\mathbf{Z}_{:,m}^+$ without involving the Cartesian product $\times$ between support latent. Essentially, our approach contrastive effectively addresses the same instance discrimination task as when considering the Cartesian product over all possible combinations. This observation aligns with insights from a disentanglement causality perspective[2] (Wang & Jordan, 2022a). Further explanations are given in § 5.3.

$$\mathcal{L}_{\text{DIOSC}} = \eta \underbrace{\sum_m \sum_{\mathcal{V}} \mathcal{D}(z_m, z_m^-)^2}_{\text{Latent-Invariant}} + \underbrace{\sum_m \sum_{\mathcal{U}} \left(1 - \mathcal{D}(z_m, z_m^+)\right)^2}_{\text{Latent-Alignment}}, \quad (4.2)$$

where $\mathcal{D}(\cdot, \cdot)$ is the cosine similarity distance, with $\mathcal{U}$ and $\mathcal{V}$ representing sets of positive and negative latent values. It is shown that both terms contribute equally to the improvement, i.e. $\eta = 1$.

---

[2]This study embraces a causal of representation learning, contrasting with DIOSC's relaxation of the independence assumption to Independence-of-Support (IoS) via contrastive.

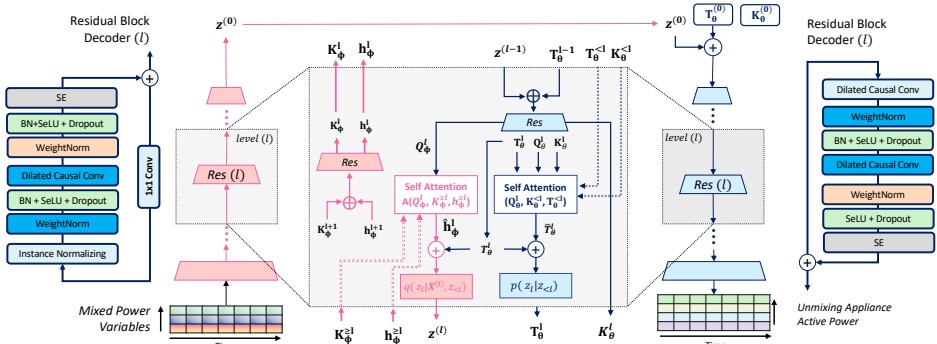

Figure 3: Performing Attentive $l$-Variational Inference entails processing power input $\mathbf{x} \in \mathbb{R}^{C \times T}$ through $l$ levels of residual blocks ($\mathbf{Res}^{(l)}$), generating key and query feature maps. Parameters $[\mathbf{K}_\phi^{(L+1)}, T_\theta^{(0)}, \mathbf{K}_\theta^{(0)}]$ are initially set to zero, and $h^{(L+1)} \stackrel{\delta}{=} \mathbf{x}$.

## 4.2 ATTENTIVE $l$-VARIATIONAL AUTO-ENCODERS

To avoid time locality during dimension reduction, and keep long-range capability we refer to an in-depth Temporal Attention with $l$-Variational layers. NVAE (Vahdat & Kautz, 2020; Apostolopoulou et al., 2021) proposed an in-depth autoencoder for which the latent space $\mathbf{z}$ is level-structured and attended locally (Apostolopoulou et al., 2021), this shows an effective results for image reconstruction. We employ Temporal Multihead Self-attention (Vaswani et al., 2017) for constructing beliefs of variational layers, allowing effective handling of long context sequences. $l$-Variational Inference is illustrated in Fig. 3, where the construction of Temporal context $\hat{T}_\theta^{(l)}$ at level $l$ relies on a preview contexts i.e $l-1$ denoted $T_\theta^{(<l)}$, query map $\mathbf{Q}_\theta^{(l)}$, and its key map $\mathbf{K}_\theta^{(<l)}$. This approach enables the model to attend to information from different representation subspaces at various scales. The use of Multihead self-attention aids in capturing diverse relationships and patterns. The detailed mechanism is given in App. D. For the remainder, we assume that DIOSC uses attentive variational inference $l$. We adopt the Gaussian residual parametrization between the prior and the posterior. The prior is given by $p(\mathbf{z}^{(l)}|\mathbf{z}^{(<l)}) = \mathcal{N}(\mu(T_\theta^l, \theta), \sigma(T_\theta^l, \theta))$. The posterior is then given by $q(\mathbf{z}^{(l)}|\mathbf{x}, \mathbf{z}^{(<l)}) = \mathcal{N}(\mu(T_\theta^l, \theta) + \delta\mu(\hat{T}_\phi^l, \phi), \sigma(T_\theta^l, \theta) \cdot \delta\sigma(\hat{T}_\phi^l, \phi))$ where $\mu(\cdot), \sigma(\cdot), \delta\mu(\cdot),$ and $\delta\sigma(\cdot)$ are transformations implemented as convolutions layers. Hence, the term $\mathcal{L}_{\mathbf{KL}}$ in Eq. 3.1 adding the residual and then the $\mathcal{L}_{\mathbf{KL}}$ is given by:

$$\mathcal{L}_{\mathbf{KL}}(\mathbf{x}; \phi, \theta) = \mathcal{L}_{\mathbf{KL}}(\mathbf{x}; \phi, \theta) + \sum_{l=1}^{L} 0.5 \times \left( \frac{\delta\mu^{(l)2}}{\sigma^{(l)2}} + (\delta\sigma^{(l)})^2 - \log(\delta\sigma^{(l)})^2 - 1 \right). \quad (4.3)$$

## 4.3 SETTING OVERALL OBJECTIVE FUNCTION

Our final objective function combines the regularization (Eq. 3.1) and the VAE loss (Eq. 3.1), which consisting of a reconstruction term $\mathcal{L}_{rec}$, a $\mathcal{L}_{\mathbf{KL}}$ term. We present balancing parameters, denoted as $\lambda$ and $\beta$, with $\lambda$ weight the disentanglement regularization and $\beta$ balancing emphasis between the reconstruction and KL divergence terms.

$$\mathcal{L}(\mathcal{D}, \phi, \theta) = \mathbb{E}_{\mathbf{X} \overset{b}{\sim} \mathcal{X}} \left[ \lambda \mathcal{L}_{\text{DIOSC}} + \frac{1}{b} \sum_{\mathbf{x} \in \mathbf{X}, \mathbf{y} \in \mathbf{Y}} \mathcal{L}_{rec}(\hat{\mathbf{y}}, \mathbf{y}; \phi, \theta) + \beta \mathcal{L}_{\mathbf{KL}}(\mathbf{x}; \phi, \theta) \right]. \quad (4.4)$$

## 4.4 HOW TO EVALUATE DISENTANGLEMENT FOR TIME SERIES?

Evaluating disentanglement in series representation is more challenging than established computer vision metrics. Existing time series methods rely on qualitative observations and predictive performance, while metrics like Mutual Information Gap (MIG) (Li et al., 2023) have limitations with continuous labels. To address this, we adapted RMIG (Carbonneau et al., 2022) for continuous labels and used DCI metrics from (Do & Tran, 2021). Our evaluation, including DCI, RMIG. The $\beta$-VAE and FactorVAE scores, can be found in App. D.6. However, these measures suffer from

| Metric | Align-axis | Unbiased | No. Condition |
|---|---|---|---|
| $\beta$-VAE | | | |
| FactorVAE | ✓ | | |
| RMIG | ✓ | | ✓ |
| SAP | ✓ | | ✓ |
| DCI | ✓ | ✓ | |
| **TDS (Ours)** | ✓ | ✓ | ✓ |

Table 1: TDS Compared to SOTA metric. (Red row the worst, Blue the best).

limitations with sequential data and do not provide measures of attribute alignment under ground truth variation. To overcome this, consider cross-correlation between latent variables $z_m$ (latent anchor) and its augmentation $z_m^+$ with respect to ground truth attribute $y_m$. Yet, practical challenges arise with multiple attributes, as this measure can be sensitive to variations within the same attribute. To address this, we introduce the compact **T**ime **D**isentanglement **S**core.

$$TDS = \frac{1}{dim(\mathbf{z})} \sum_{n \neq m} \sum_{k} \frac{||z_m - z_{n,k}^+||^2}{\text{Var}[z_m]}, \tag{4.5}$$

where $z_{n,k}^+$ is an augmentation of $z_m$, and $\text{Var}[z_m]$ variance of $z_m$. When a set of latent variables are not axis-aligned, each variable can contain a decent amount of information regarding two or more attributes. A wide gap between unaligned variables indicates an entanglement. TDS excels in axis alignment (c.f. Table. 1), is unbiased across hyperparameters.

## 5 EXPERIMENTS

### 5.1 ARCHITECTURE SETTINGS AND DATA AUGMENTATION FRAMEWORK

**Residual Blocs.** We enhance our Residual model by replacing traditional components in residual blocks with Sigmoid Linear Units (SiLU) (Elfwing et al., 2017). SiLU offers advantages such as faster training, robust feature learning, and superior performance compared to weight normalization.

Our framework is given in Fig. 3, we set $L = 16$, and we fix an time window input to 256 steps, for the latent space dimension we fix $d_z = 16$.

**Squeeze-and-Excitation on Spatial and Temporal:** The SE block improves neural networks by selectively emphasizing important features and suppressing less relevant ones. Extending SE for time series data enhances the capture of significant temporal patterns in sequences.

| DIOSC($L=8$) | KL $\downarrow$ | RMSE $\downarrow$ | Time (s) $\downarrow$ |
|---|---|---|---|
| ReLU | 0.734 | 0.734 | 28800 |
| SiLU | **0.671** | **0.671** | 21600 |
| ReLU+SE | 0.721 | 0.721 | 32760 |
| SiLU+SE | **0.582** | **0.582** | **23040** |

Table 2: Metrics on UK-Dale ($\downarrow$ lower is better, $\uparrow$ higher is better Top-2 , Top-1 ).

**Pipeline Augmentation for Electric Load Monitoring.** Four augmentations were sequentially applied to all contrastive methods' pipeline branches. The parameters from the random search are: 1) **Crop and delay:** applied with a $0.5$ probability and a minimum size of $50\%$ of the initial sequence. 2) **Cutout or Masking:** time cutout of 5 steps with a $0.8$ probability. 3) **Channel Masks powers:** each power (reactive, active, and apparent) is randomly masked out with a $0.4$ probability. 4) **Gaussian noise:** random Gaussian noise is added to window activation $y_m$ and $\mathbf{x}$ with a standard deviation of $0.1$. The impact of each increase is detailed in the Appendix. D.2.

**Pipeline Correlated Sampling Attributes.** We evaluate the model's robustness to data correlations by examining various pairs, primarily focusing on linear correlations between two appliances and scenarios where one device correlates with two others. For this, we parameterize these correlations by sampling from a common distribution $p(y_1, y_2) \propto \exp\left(-||y_1 - \alpha y_2||^2/2\sigma^2\right)$, where $\alpha$ is a scaling factor, and $\sigma$ indicates the strength of the correlation. We extends the (Träuble et al., 2021) framework beyond time series and adapts it to cover correlations between multiple devices operating in a $T$ time window. Scenario examples include: **No Correlation** ($\sigma = \infty$); **Pair: 1** (clothes dryer/oven, $\sigma = 0.3$); **Pair: 2** (washing machine/dishwasher, $\sigma = 0.4$), and **Random pair** (randomly selected pairs, $\sigma = 0.8$). Additional correlation pairs are detailed in Appendix. D.3.

### 5.2 EXPERIMENTAL SETUP

**Datasets.** We conducted experiments on three public datasets: UK-DALE (Kelly & Knottenbelt, 2015), REDD (Kolter & Johnson, 2011), and REFIT (Murray et al., 2017) providing power measurements from multiple homes. We focus on six appliances: Washing Machine, Oven, Dishwasher, Cloth Dryer, Fridge. We performed cross-tests on different dataset scenarios, each with varying sample sizes. Specifically, scenario **A** involved training on REFIT and testing on UK-DALE, $18.3k$ samples with time window $T = 256$, and frequencey of 60Hz, the test set consisted of $3.5k$ samples, scenario **B** involved training on UK-DALE and testing on REFIT with $13.3k$ samples, and scenario **C** involved training on REFIT and testing on REDD with $9.3k$ samples. The augmentation pipeline is applied for all scenarios. For training and testing under correlation, we use the corresponding sampling.

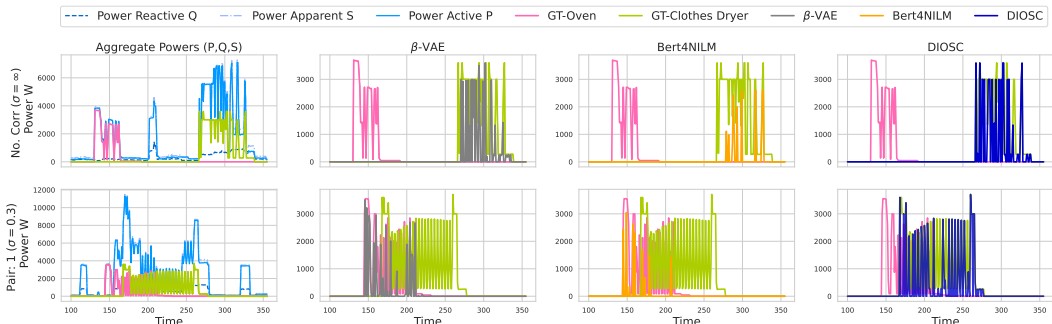

Figure 4: Prediction Clothes dryer under in correlated case (top) and uncorrelated case (bottom) over a time window of 256min. Moving from left to right, the graph illustrates the aggregated power (**P,Q,S**) alongside the ground-truth **Clothes-dryer** to be identified under correlation with **Oven**.

**Baseline and Evaluation.** We compare `DIOSC` with downstream task models in energy, Bert4NILM (Yue et al., 2020) as a baseline and S2P (Yang et al., 2021), S2S (Chen et al., 2018a), for those model we keep the same configuration as the original implémentation. We provide also variant $\beta$-TC/Factor/-VAE implemented for time series, compared to `D3VAE` (Li et al., 2023) and `NVAE` (Vahdat & Kautz, 2021), and RNN-VAE (Chung et al., 2015). We compare these model using RMSE, and we compute disentanglement metrics: RMIG, DCI, TDS. The metrics have been evaluated by both, either sampling from the correlated data or from the uncorrelated distribution.

**Experimental Platform.** We conduct 5 seeds of experiments, reporting the averaged results and standard deviation. Based on the grid search, we found that `DIOSC`'s best performance is obtained by ($\lambda = 2.3$, $\beta = 1.5$). The experiments are performed on four NVIDIA A100 GPUs. Hyperparameter settings are available in Appendix D.

## 5.3 Performance and Informativity of Contrastive

*Finding: `DIOSC` performs better in Out-of-distribution (under correlated data).*

To assess `DIOSC`'s regularization robustness to correlated appliances, we examine scenarios involving pairs defined in § 5.1. From Fig. 6 the increased disentanglement through `DIOSC` gives consistent improvements in all cases, and gets more pronounced in the low data regime, indicating higher sample efficiency, as expected from better disentanglement even the correlated in pairs. Fig. 3 show the regression results as we see even when signals are correlated the disentangling is acheived and relative improvements up to +20% in RMSE. We find factorization of supports using `DIOSC` on the training data is strongly relate to downstream disentanglement even when experiencing a strong correlation during training.

| Sc. | Methods | No Corr $\sigma = \infty$ | | | Pairs: 1 $\sigma = 0.3$ | | | Pairs: 2 $\sigma = 0.4$ | | | Random Pair $\sigma = 0.8$ | | |
|---|---|---|---|---|---|---|---|---|---|---|---|---|---|
| | Metrics $-\downarrow$ | DCI ↓ | TDS ↓ | RMSE ↓ | DCI ↓ | TDS ↓ | RMSE ↓ | DCI ↓ | TDS ↓ | RMSE ↓ | DCI ↓ | TDS ↓ | RMSE ↓ |
| A | Bert4NILM | - | - | 56.4 ±2.58 | - | - | 70.2±1.45 | - | - | 70.92±1.15 | - | - | 70.92±1.15 |
| | S2S | - | - | 54.3 ±3.12 | - | - | 69.5±3.56 | - | - | 72.31±2.45 | - | - | 69.95±3.26 |
| | $\beta$-VAE | 72.4±3.10 | 0.96±.15 | 48.6 ±2.32 | 72.4±3.10 | 0.96±.15 | 52.6 ±2.31 | 72.4±3.10 | 0.96±.15 | 54.73±1.54 | 74.29±2.04 | 1.08±.09 | 52.99±1.91 |
| | $\beta$-TCVAE | 78.0±1.09 | 0.94±.13 | 43.2 ±2.23 | 78.0±1.09 | 0.94±.13 | 49.2 ±1.13 | 77.23±0.76 | 0.94±.13 | 50.87±1.17 | 79.74±0.84 | 1.07±.11 | 49.65±1.43 |
| | FactorVAE | 68.4±2.41 | 0.97±.03 | 47.7 ±1.35 | 68.4±2.41 | 0.97±.03 | 53.2 ±1.02 | 69.78±1.43 | 0.97±.03 | 54.32±0.64 | 69.95±1.63 | 1.00±.02 | 53.45±0.82 |
| | HFS | 79.8 ± .10 | 0.64 ± .05 | 57.2 ± 2.15 | 79.8 ± .10 | 0.64 ± .05 | 61.3 ± 1.82 | 79.56±0.28 | 0.64±.05 | 62.33±1.23 | 80.37±.05 | 0.72±.03 | 61.64±1.52 |
| | $\beta$-VAE + HFS | 73.1±1.01 | 0.69±.02 | 34.4±1.89 | 73.1±1.01 | 0.69±.02 | 38.1±1.34 | 73.59±0.86 | 0.69±.04 | 39.65±0.87 | 74.25±0.59 | 0.73±.05 | 38.48±1.04 |
| | $\beta$-TCVAE + HFS | 67.2±2.01 | 0.52±.02 | 24.3 ±1.81 | 67.2±2.01 | 0.52±.02 | 27.4 ±1.13 | 67.51±1.84 | 0.52±.07 | 28.94±0.66 | 68.79±1.27 | 0.58±.04 | 27.77±0.83 |
| | DIOSC | 63.5±1.35 | 0.49±.02 | 19.6±1.95 | 69.3±1.2 | 0.4±.02 | 22.3±1.79 | 70.3±0.82 | 0.49±.02 | 23.97±1.19 | 67.12±0.91 | 0.51±.01 | 22.63±1.49 |
| B | Bert4NILM | - | - | 57.85 ±1.88 | - | - | 68.8±1.12 | - | - | 73.41±1.35 | - | - | 72.78±0.88 |
| | S2S | - | - | 56.38 ±2.22 | - | - | 67.8±2.76 | - | - | 73.95±1.91 | - | - | 70.92±2.25 |
| | $\beta$-VAE | 73.78±2.68 | 1.08±.09 | 50.14 ±1.87 | 75.47±1.98 | 0.82±.10 | 51.7±1.79 | 70.8±2.62 | 0.85±.11 | 55.98±1.27 | 76.18±1.54 | 1.16±.08 | 54.83±1.58 |
| | $\beta$-TCVAE | 79.57±0.84 | 1.07±.11 | 45.72 ±1.68 | 80.23±0.54 | 0.81±.09 | 48.3±0.94 | 76.2±0.54 | 0.83±.10 | 51.74±0.94 | 80.88±0.53 | 1.15±.10 | 51.15±1.10 |
| | FactorVAE | 70.14±1.89 | 1.00±.02 | 49.02 ±1.05 | 71.89±1.24 | 0.94±.02 | 52.4±0.85 | 68.7±1.13 | 0.92±.02 | 55.24±0.42 | 71.57±1.27 | 1.06±.01 | 54.68±0.64 |
| | HFS | 80.12±.05 | 0.72±.03 | 58.49 ±1.45 | 80.26±.03 | 0.56±.03 | 6.0±1.42 | 78.8±0.15 | 0.58±.03 | 63.79±0.97 | 80.61±.02 | 0.80±.02 | 63.22±1.17 |
| | $\beta$-VAE + HFS | 74.47±0.61 | 0.73±.05 | 36.09±1.25 | 75.12±0.41 | 0.67±.02 | 37.4±1.04 | 72.8±0.52 | 0.64±.03 | 40.92±0.66 | 75.07±0.43 | 0.75±.03 | 39.68±0.80 |
| | $\beta$-TCVAE + HFS | 68.54±1.36 | 0.58±.04 | 25.88 ±1.20 | 69.28±1.01 | 0.46±.01 | 26.7±0.88 | 66.7±1.51 | 0.45±.02 | 29.82±0.51 | 7.04±0.93 | 0.72±.02 | 40.49±0.64 |
| | DIOSC | 64.42±0.96 | 0.51±.01 | 21.35 ±1.80 | 65.11±0.66 | 0.39±.01 | 21.5±1.44 | 69.5±0.43 | 0.48±.01 | 24.94±0.87 | 65.05±0.71 | 0.55±.01 | 24.05±1.30 |
| C | Bert4NILM | - | - | 58.29 ±2.16 | - | - | 75.6±1.68 | - | - | 71.73±1.66 | - | - | 76.05±1.87 |
| | S2S | - | - | 56.28 ±2.43 | - | - | 73.8±3.91 | - | - | 74.76±3.75 | - | - | 73.47±4.12 |
| | $\beta$-VAE | 74.17±2.01 | 1.03±.09 | 50.18 ±1.92 | 73.84±1.56 | 0.72±.12 | 55.7±2.47 | 76.1±3.36 | 1.07±.17 | 56.32±2.31 | 73.95±1.93 | 1.16±0.11 | 55.90±2.40 |
| | $\beta$-TCVAE | 79.21±0.89 | 0.98±.10 | 45.11 ±2.03 | 79.48±0.75 | 0.78±.08 | 50.9±1.27 | 78.85±0.94 | 1.05±.15 | 51.19±1.84 | 80.57±0.95 | 1.10±0.11 | 51.17±1.85 |
| | FactorVAE | 70.23±1.70 | 0.99±.02 | 49.12 ±1.18 | 69.75±1.53 | 0.99±.03 | 56.4±1.11 | 70.92±1.58 | 0.99±.05 | 55.48±1.25 | 70.43±1.74 | 1.05±.02 | 54.61±1.34 |
| | HFS | 8.04±.06 | 0.67 ± .03 | 59.04 ±1.74 | 80.11±.05 | 0.60±.04 | 62.9±1.98 | 79.91±0.36 | 0.69±.07 | 63.52±1.94 | 80.42±.06 | 0.73±.03 | 63.83±2.01 |
| | $\beta$-VAE + HFS | 69.03±0.79 | 0.70±.01 | 35.65±1.59 | 74.14±0.82 | 0.74±.01 | 40.5±1.49 | 74.26±0.95 | 0.71±.06 | 40.32±1.38 | 74.84±0.51 | 0.78±.05 | 39.38±1.19 |
| | $\beta$-TCVAE + HFS | 69.04±1.45 | 0.54±.01 | 25.85 ±1.45 | 68.37±1.31 | 0.47±.01 | 28.9±1.28 | 69.07±2.02 | 0.59±.09 | 30.38±1.24 | 69.84±1.43 | 0.62±.04 | 29.29±1.13 |
| | DIOSC | 64.87±1.07 | 0.50±.01 | 19.6±1.95 | 70.54±0.60 | 0.50±.01 | 21.1±1.92 | 71.2±0.94 | 0.44±.03 | 26.97±1.04 | 67.72±1.01 | 0.57±0.01 | 24.12±1.58 |

Table 3: Average scores `DCI`, `TDS`, and RMSE vary from No. Correlation (left) to every appliance correlated with one confounder (right) on uncorrelated test data. Red to blue, with bold indicating the best performance per correlation. (↓ lower is better, ↑ higher is better  Top-2 , Top-1 ).

# 6 ABLATION STUDIES

## 6.1 DIOSC PRESERVES ITS ROBUSTNESS IN CORRELATED SCENARIOS

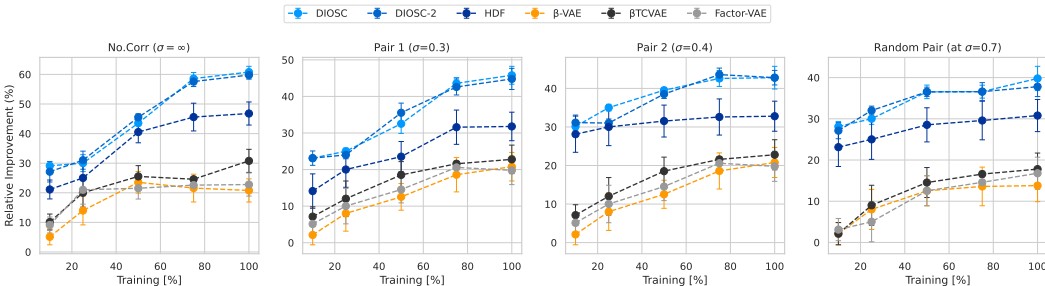

Figure 5: Relative RMSE (%) improvement over Bert4NILM for six devices using DIOSC, $\beta$-VAE, and FactorVAE, with the amount of labelled training data as a variable parameter.

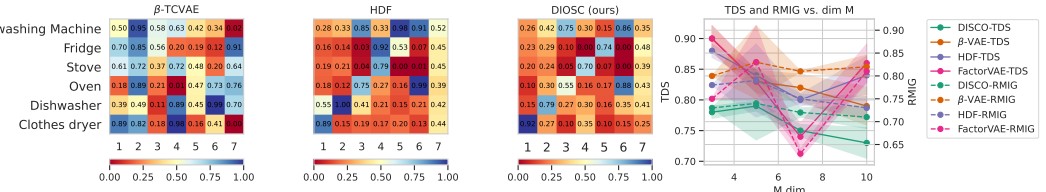

Figure 6: We find strong correlation between DIOSC and disentanglement metrics vary with $M$ (right), linked to classifcation accuracy of each compenents $z_m$ on $y_m$ labeled test data (Left), Darker Blue $\rightarrow$ High accuracy.

*Finding: DIOSC preserves its robustness in correlated scenarios and achieves comparable performance to baseline models with less training sample data.*

Training with the same $l$ variational inference model with the different regularisation variants results shows that DIOSC outperforms SOTA as shown in Fig. 5, mainly in the uncorrelated cases with only 50% of labelled data, which corresponds to HDF (Roth et al., 2023). With 80% of data, DIOSC scores 14% better than HDF and 61.4% better than the baseline Bert4NILM. In the correlated scenarios (pair 1 and 2), $\beta$/Factor/TC-VAE shows weaker performance, while DIOSC consistently outperforms HDF and the baseline.

## 6.2 IN-DEPTH SELF-ATTENTION $l$-VAEs LEARN AN EFFECTIVE REPRESENTATION.

*Finding: As DIOSC deepens, representation increases over 20% (40% in TDS), downtasking boosts performance, and attention mechanisms contribute a 10% improvement.*

In Table 7, we employ $l$-Variational Inference with the DIOSC regularizer, both with and without self-attention, and explore its application with alternative structures tailored for time series, particularly those residual in D3VAE. Our observations reveal two key findings. Firstly, incorporating DIOSC with another regularization method slightly enhances results, as the alternative regularizer assumes independent factoriza-

| Method | Depth ($L$) | NRMSE ↓ | RMIG ↓ | TDS ↓ |
|---|---|---|---|---|
| VAE (baseline) | - | 0.928 | 0.921 | 0.935 |
| VAE (baseline)+DIOSC | - | 0.929 | 0.924 | 0.931 |
| FactorVAE | - | 0.942 | 0.931 | 0.973 |
| $\beta$-TCVAE | - | 0.931 | 0.918 | 0.937 |
| $\beta$-TCVAE+DIOSC | - | 0.930 | 0.922 | 0.933 |
| DIP-VAE | - | 0.932 | 0.915 | 0.939 |
| DIP-VAE+DIOSC | - | 0.928 | 0.926 | 0.930 |
| DIOSC | 8 | 0.50 | 0.73 | 0.71 |
| DIOSC w/o Attention | 8 | 0.54 | **0.71** | 0.72 |
| DIOSC | 16 | **0.49** | 0.74 | 0.70 |
| DIOSC w/o Attention | 16 | 0.52 | **0.72** | 0.73 |
| DIOSC | 32 | **0.48** | 0.75 | **0.69** |

Table 4: Average Normalized RMSE, RMIG, and TDS Scores for Variants DIOSC w/,w/o Attention, as $L$ Increases. (↓ lower is better, ↑ higher is better  Top-2 , Top-1 ).

tion, potentially compromising the relaxing effect. Secondly, DIOSC demonstrates improved performance with increasing values of $L$, and the TDS correlates positively with performance, while RMIG suggests that using DIOSC with attention leads to well-disentangled representations. Notably, the attention mechanism proves efficient by enhancing the overall model performance.

## 6.3 ROBUSTNESS, DISENTANGLEMENT, AND STRONG GENERALIZATION

*Finding: DIOSC demonstrates robust disentanglement performance across varying dimensions, while FactorVAE exhibits degradation as dimensionality increases $M \uparrow$.*

In Fig. 6 (right), we report the disentanglement performance of `DIOSC` and FactorVAE on the Uk-dale dataset as $M$ is increased. FactorVAE (Higgins et al., 2016) is the closest TC-based method it uses a single monolithic Discriminator and the density-ratio trick to explicitly approximate TC. Computing $TC(\mathbf{z})$ is challenging to compute as $M$ increases. The results for $M = 10$ (scalable $\approx \times 3$) are included for comparison. The average disentanglement scores for `DIOSC` $M = 7$ and $M = 10$ are lower and very close, indicating that its performance is robust in $M$. This is not the case for HDF Factor/$\beta$-VAE. It performs worse on all metrics when m increases. Interestingly, HDF $M = 7$ seems to recover its performance on most metrics. Despite this, the difference suggests that HDF and Factor/$\beta$-VAE are not robust to changes in $M$. The optimal $M$ for HDF and TC/$\beta$-VAE, shown in Fig. 6 (left), indicates promising accuracy for HDF, despite being no better than `DIOSC`.

## 7 Discussion and Conclusion

To address the limitation of assuming independence in existing time series disentanglement methods, which may not accurately reflect real-world correlated data, our approach focuses on recovering correlated data. By relaxing the independence factorization assumption to independence-of-support via contrastive learning, our method achieves identification and disentanglement, enabling the model to encode attribute variability in the latent space. `DIOSC`, combines contrastive regularization and $l$-Variational Autoencoder for time series, we demonstrate that promoting pairwise factorized support is sufficient for disentangling time series. Consistent with our theoretical findings, `DIOSC` outperforms baseline methods by more than **+61.4%** in downstream tasks of NILM across datasets with various correlation shifts, highlighting the benefits of enhanced disentanglement for out-of-distribution generalization in representation learning. Future work should explore support factorization for time series with causal notions rather than independence.

**Limitations of Theory.** Although we posit that our theoretical assumptions encapsulate crucial aspects of time series representation learning under strong correlation, they may be subject to varying degrees of violation in practical scenarios characterized by correlations. For instance, the relaxation assumption to IOS (Asm. 4.1) regarding the batch is influenced by its size, which we consider as a hyperparameter during training.

## 8 Broader Impact

Our proposed method enables effective representation learning for time series data related to energy load, offering broad applicability across various downstream tasks. In this context, we showcase its efficacy in scenarios characterized by strong correlations. The scalability of our approach, particularly when applied to scaled versions featuring a large number of appliances, facilitates its generalization across domains, establishing foundational models for energy disaggregation. The potential societal benefits, such as enabling household consumption determination, are particularly notable within the context of smart grid systems. As evidenced by its successful implementation in smart grid management, our method readily adapts to efficiently detect appliance consumption patterns. This capability not only aids in energy management but also provides users with valuable feedback regarding optimal utilization during off-peak hours, thereby optimizing energy consumption and consequently reducing carbon footprint. Such contributions underscore the significant societal and environmental advantages afforded by AI-driven models.

## 9 Acknowledgements

This work was granted access to the HPC resources of IDRIS under the allocation AD011014921 made by GENCI (Grand Equipement National de Calcul Intensif). Part of this work was funded by the TotalEnergies Individual Fellowship through One Tech. Special appreciation is given to Thierry Luci, head of the Applied Scientist AI Team at One Tech, for his leadership and support, and to the team for their active participation in insightful discussions.

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

# Supplementary Material

To facilitate a comprehensive examination of our paper, we present additional results and furnish complete proofs for the assumptions articulated in the main manuscript. This supplementary material is meticulously organized as follows:

## Table of Contents

