# OpenReview forum: "Disentangling Time Series Representations via Contrastive Independence-of-Support on l-Variational Inference"
_ICLR.cc/2024/Conference — ICLR 2024 poster_

### Official Review · Reviewer_cDdv · 2023-10-17

**Soundness:** 4 excellent
**Presentation:** 4 excellent
**Contribution:** 4 excellent
**Rating:** 8
**Confidence:** 5

**Summary:**

The paper discusses the importance of learning disentangled representations for Time Series data, specifically in the context of home appliance electricity usage. The goal is to enable users to better understand and optimize their energy consumption, thereby reducing their carbon footprint. The authors frame the problem as one of disentangling the role of each appliance (e.g., dishwashers, fridges) in total electricity usage.

Unlike existing methods that assume attributes (appliances in this case) operate independently, this work acknowledges that real-world time series data often show correlations between attributes. For instance, dishwashers and washing machines might be more likely to operate simultaneously during the winter season.

To address these challenges, the authors propose a method called DisCo (Disentangling via Contrastive), which employs weakly supervised contrastive disentanglement. This approach allows the model to generalize its representations across various correlated scenarios and even to new households. The method incorporates novel VAE layers equipped with self-attention mechanisms to effectively tackle temporal dependencies in the data.

To evaluate the quality of disentanglement, the authors introduce a new metric called TDS (Time Disentangling Score). The TDS proves to be a reliable measure for gauging the effectiveness of time series representation disentanglement, thereby making it a valuable tool for evaluation in this domain.

Overall, the paper argues that disentangled representations, particularly those achieved using their DisCo method, can enhance the performance in tasks like reconstructing individual appliance electricity consumption.

**Strengths:**

The method is very sound with mathmaticaly correct derivations.
The addressed problem of disentagling latent factors in VAE type of models is very important.
Specifically, he paper addresses the unrealistic assumption of independence among generative attributes that is often present in traditional untangling methods. In contrast to these traditional approaches, DisCo focuses on recovering correlated data by encoding a wide range of possible combinations of generative attributes in the learned latent space.

The authors assert that simply encouraging pairwise factorized support in the latent space is sufficient for achieving effective disentanglement, even when data attributes are correlated. This is an important finding.

In terms of performance, DisCo is shown to be competitive with downstream task methods, exhibiting significant improvements of over +60% across a variety of benchmarks in three different datasets undergoing correlation shifts (Finding 5.1). This is a strong aspect of the work.

Additionally, the capability of DisCo to adapt across correlation shifts leads to better out-of-distribution generalization, especially when these shifts are more severe. This fulfills one of the key promises of learning disentangled representations, which is to improve the model's robustness and generalizability.

**Weaknesses:**

I enjoyed the paper and did not find important weaknesses.

**Questions:**

Please discuss how sensitive the method is to hyperparameter selection.

---

> ### Author Response · Authors · 2023-11-21
> **Thank you for your review**
>
> We appreciate your thorough review and insightful feedback on our paper. The suggestions and concerns raised have been duly noted, and we have made significant updates to address them, we will update the current version during this discussion. Major changes in the paper from the previous version will be highlighted in ***gray color***.
>
> Thank you for acknowledging the strengths of our paper in tackling a challenging problem. We note that additional experiments were conducted during the review phase, and our code repos are updated with more ablation studies. This establishes itself as the inaugural framework for disentangling time series in a library.
>
> ### Modifications
>
> - **Sensitivity Analysis:** We've integrated supplementary experimental findings and engaged in a thorough discussion covering all aspects of the training process. The forthcoming version will include corresponding adjustments to the codebase, ensuring transparency and reproducibility.
>
> - **Hyperparameter Selection Discussion:** We have included a detailed discussion on our parameter choices, addressing the sensitivity of our method to data and its correlations.
>
> We believe these enhancements contribute significantly to the overall quality and completeness of our work. Your valuable input has been instrumental in refining our paper, and we are grateful for your time and expertise.
>
> Should you have any further comments or queries, please feel free to let us know. We look forward to your continued guidance.
>
> Best,
>
> Authors,

---

> > ### Author Response · Authors · 2023-11-23
> > **Updates to our rebuttal**
> >
> > Dear Reviewer cDdv,
> >
> > We trust this message finds you well. Your feedback has played a crucial role in improving the analysis of how sensitive the method is to hyperparameters, and we sincerely appreciate your time and attention.
> >
> > We've carefully incorporated your suggestions in both the main paper and appendix, aiming to meet your expectations with this revised version. Thank you for your valuable input, and we appreciate your time and consideration.
> >
> > Best regards,
> >
> > Authors,

---

### Official Review · Reviewer_jyEy · 2023-10-27

**Soundness:** 3 good
**Presentation:** 3 good
**Contribution:** 3 good
**Rating:** 8
**Confidence:** 4

**Summary:**

This study explores disentangled representations for time series data, with a primary emphasis on achieving representation generalization across diverse, interrelated scenarios. They focused on a specific application of electric load monitoring application where computing different household appliances contribution in a total load is the task.
In the context of Variational Autoencoders (VAE), this study draws inspiration from Roth et al., 2023, who addressed correlated attributes in an image processing context by replacing the independence constraint over attributes in the latent space (by a regularization term of the Kullback-Leibler divergence between the posterior of the latent attributes and a standard Guassian distribution), with the Hausdorff Factorized Support (HFS) assumption. The authors have adapted this idea for time series data and introduced the use of cosine similarity instead. Consequently, this approach no longer necessitates independent latent activations for different appliances.
The main idea is to address appliance correlations with weakly supervised contrastive disentanglement, promoting similarity for the same appliances and dissimilarity for absent appliances in latent representations. This is achieved through a loss function composed of two terms, one for alignment based on correlation and another to minimize redundancy between latent variables.
In addition, the authors proposed l-variational inference layers with self-attention mechanism to address temporal dependencies. Additionally they propose a metric of  Time Disentangling Score (TDS) to evaluate the  disentanglement performance in time series data.

**Strengths:**

The paper presents several intriguing novelties.
1) Using pairwise similarity rather than independence assumption in VAE, to consider the correlated representations.
2) l-variational inference layers with self-attention mechanism
3) A metric of  Time Disentangling Score (TDS) to evaluate the  disentanglement performance in time series data

 The authors have tackled a captivating problem, successfully adapting image processing techniques to the more complex domain of time series data.

**Weaknesses:**

The paper needs some modifications to make it easier to read (some suggestions given in the Questions).
The experimental results are very abstract (some suggestions in Questions part)
The application worth more explanation, the description lacks either an illustration or it is abstract.

**Questions:**

-Using cosine similarity instead of HFS needs more elaboration.

-Section 3.2 would benefit from a dedicated illustration demonstrating ATTENTIVE l-VARIATIONAL AUTO-ENCODERS, along with the corresponding notations used in the text.

-The authors have effectively presented the formulation for the usecase; however, in the experimental results, which I find somewhat abstract, there's a lack of a specific example illustrating how X and Y values for a time window are displayed, along with different rows of Y, etc.

-In section 4.1, should be included how exactly augmentation is performed and how many, it is very abstract now.

-In Section 2, specifically concerning contrastive learning, the evaluation of appliance dissimilarity in "x" and "x-" is not explicitly clarified. Is labeling used for this purpose? What if the appliances are not the exact same but should exhibit similar behavior? How are such cases addressed? Additionally, the preparation of negative and positive samples is not detailed. Have you considered ensuring that there are no common or similar appliances in these two sets, and if so, how was this determined? Providing further explanation or an illustration could enhance the clarity of data preparation, which is a crucial aspect of the methodology.
-How many training examples did you use for linking?
“We link the learned latent representation to ground-truth attributes using a limited number of pair labels”
-After equation 2, In this test, the latent variable is represented as "z," which is defined as a matrix of dimensions (M + K) × dz, where "K" and "dz" should be introduced and define.

---

> ### Author Response · Authors · 2023-11-21
> **Thank you for your review - More additional experiments have been added with explanations.**
>
> We appreciate the thorough review and constructive feedback on our paper. The suggestions and concerns raised have been duly noted, and we have made significant updates to address them, we will update the current version during this discussion. Major changes in the paper from the previous version will be highlighted in ***gray color***.
>
> Thank you for acknowledging the strengths of our paper in tackling a challenging problem. We emphasize these contributions more explicitly in the revised version.
>
> We note that additional experiments were conducted during the review phase, and our code repos are updated with more ablation studies. This establishes itself as the inaugural framework for disentangling time series in a library.
>
> ### Weaknesses and Modifications
>
> We appreciate the feedback regarding the readability and abstract nature of the experimental results and application description. We have addressed these concerns by providing additional details and clarification in the revised version (which will be updated during this discussion). Specific modifications have been made based on the suggestions provided in the Questions section.
>
> #### **[A/Q]**
>
> 1.   We have provided a more elaborate more explanation for Hausdaurff Factorization support (HFS) in the revised version.
>
> 2. Section 3.2 is moved to 4.2 and now includes a dedicated illustration (Figure. 3) demonstrating Attentive l-Varitiaonla Autoencodeur along with corresponding notations.
>
> 3. Experimental results section now includes specific examples illustrating how $\textbf{x}$ and $\textbf{y}$ values (see section 5, other results in the appendix) for a time window are displayed for each experiment, addressing concerns about abstractness we add the figure of mixed signal input and unmixed results (prediction).
>
> 4. **Negative and Positive Samples.** We define a positive sample $\textbf{x}$ as a mixed power state where an appliance \(m\) is activated (\(y_m > 0\)), and `\(\textbf{x}^{+}\)` represents its augmentation. For this setup, labeled data is required. Finding a negative sample is comparatively straightforward, as any other sample $\textbf{x}$ can be chosen where $m$ is not activated $y_{m}=0$, but at least one other appliance is activated (no need for their labels). Augmentations (see our answer below) of both positive and negative samples generate sets for positives and negatives for appliance $m$. This process is repeated for $M$ appliances as preparation data.
>
> 5. Given access to labels for each appliance, we meticulously consider the correlation and similarity among appliances, assessing whether they exhibit similarities, correlations, or are uncorrelated. In response to your feedback, we elaborate on this aspect in the revised version and explicitly specify the number of samples utilized for training.
>
> 6. **Time-based Augmentation*. In Section 5.1 experiments, we have provided detailed insights into the augmentation process to enhance clarity. Additional algorithmic details are available in the appendix. The four augmentations were sequentially applied to both negative and positive: 1). **Crop and delay**: crop and delayed in time by 5 steps with a probability of 0.5. 2). **Channel Masks powers**: Randomly mask out each power channel (reactive, active, and apparent) with a probability of 0.4. 3). **Cutout or Masking**: Time cutout of 5 steps with a probability of 0.8. 4). **Add Gaussian Noise**: Introduces random Gaussian noise to window activation  $y_{m}$ and $\mathbf{x}_{m}$ with a standard deviation of 0.1. We add a comparison of performance between different choices of augmentation.
>
> 6. Section.3 now provides explicit clarification on the evaluation of appliance dissimilarity in $\mathbf{x}$ and $\mathbf{x}^{+}$ addressing concerns about labeling and similar behavior. Further details on the preparation of negative and positive samples have been included, along with an illustration to enhance clarity.
>
> 7. After Equation. 2, we have introduced and defined $K$ and $d_{z}$ fixed before training ($K$ unknowing appliance to find) in the revised manuscript.
>
> 8. We favor Cosine Similarity over alternative distance measures based on specific considerations within our context. Our choice is motivated by the computational efficiency of Cosine Similarity compared to Earth Mover's Distance (EMD), as outlined in our study detailed in Appendix C. Notably, we observed that Cosine Similarity exhibits scale-invariance, focusing on direction rather than being affected by the magnitude of the latent. In contrast, EMD incorporates transportation costs between distributions, introducing a sensitivity to scale.
>
> We value your insightful review and remain committed to addressing all raised concerns diligently. Should you have further inquiries about our paper or its underlying assumptions, we are more than happy to provide detailed responses.
>
> Best,
>
> Authors,

---

> > ### Comment · Reviewer_jyEy · 2023-11-22
> >
> > Thanks for further clarifications and your proposed modifications can improve the paper readability. However, when I checked the manuscript it  still does not contain the modification you mentioned.

---

> ### Author Response · Authors · 2023-11-22
>
> Sorry there was a slight problem of pdf upload. Could you please check the last pdf we uploaded.  Please if you have any further questions about our paper or its underlying assumptions, we will be happy to provide detailed answers.
>
> Best,
>
> Authors,

---

> > ### Author Response · Authors · 2023-11-23
> > **Updates to our rebuttal ! Any Aditional question**
> >
> > Dear Reviewer jyEy22,
> >
> > We hope this message finds you well. Your feedback has been invaluable in enhancing our paper. Thank you for your time and attention. We have incorporated changes based on your suggestions, and we hope this version meets your expectations. We appreciate your valuable input.
> >
> > Thank you for your time and consideration.
> >
> > Best regards,
> >
> > Authors,

---

### Official Review · Reviewer_a2rd · 2023-11-01

**Soundness:** 1 poor
**Presentation:** 1 poor
**Contribution:** 2 fair
**Rating:** 1
**Confidence:** 3

**Summary:**

The paper seems to be an application of disentangled representation learning for home appliance electricity usage. The authors propose to combine contrastive and variational losses. Unfortunately, the paper falls somewhere between methodological novelty and application, making it difficult to understand where the main contributions of the paper will lie. In general, I found the paper very hard to read.

**Strengths:**

The main strength of the paper is its approach to tackling the important problem of disentangled representation learning, which may contribute to reducing carbon footprint.

**Weaknesses:**

The paper lacks proper organization and has a tendency to include some unproven (or wrong) claims. For instance, in the introduction, the authors mention that "Disentanglement via VAE **can be achieved** by a regularization term of the Kullback-Leibler divergence[]," which is not necessarily true without certain strong assumptions and underlying conditions. The Beta-VAE paper has some qualitative evidence showing how the images are more disentangled compared to VAE. The authors also claim, "Rather than training separate auto-encoders for individual appliances," which requires empirical validation with proper citations.

In addition to these issues, the notations used are very confusing and are not defined before they are referenced. For example, in the proposed method section, the notation $z_m^+$ is used without description it. It is also unclear how it differs from $\bf{z}$ or $z$.

The main goal of this paper remains unclear to me. For example, the authors mentioned, "The primary goals of this work are twofold: to effectively address the NILM problem and to obtain a disentangled representation of input data." However, it is unclear what the NILM problem is, what the nature of the input data is, and how the authors plan to achieve a disentangled representation that distinguishes itself from previous works. One issue might be that the problem statement and preliminaries are somewhat intertwined.

The color meaning used in the tables of result section is not clear. Even it is not clear how TDS (as a metric) has been compared with VAE and Beta-VAE in Table 1.

**Questions:**

- I strongly suggest the authors make their main contributions clear at the end of the introduction.

- There is no related work section.

---

> ### Author Response · Authors · 2023-11-22
> **Thank you for your reply - Some Clarification about  Disentanglement and Independence-of-Support Factorization via Contrastive**
>
> We regret the delay in updating the version due to an error on our part in the PDF upload and any resulting confusion about the paper's name. We have now rectified the situation by updating everything, including the separate appendix.
>
> To avoid any potential confusion, we have revised the paper's name from Disco to **Diosc**: Disentanglement and Independence-of-Support Factorization via Contrastive.  ***We have also been notified that other works [2,3], far from our paper, coincidentally bear the same name***.  In response to your question, our paper is distinct from [1], and the only common characteristic is both use contrastive methods. We have also modified our citation to acknowledge [1] in related work of our updated version. We clarify the difference below:
>
> 1. In [1], the focus is on exploiting in an unsupervised manner the pre-trained generative models non-disentangled such as VAE, and GAN by navigating latent space, specifically using it on a pre-trained GAN. In contrast, our work emphasizes enforcing disentanglement during the training phase of VAE itself, by relaxing the assumption of attribute independence factorization to independence-of-support (refer to assumption 4.1 in our paper).
>
> 2. The use of a pre-trained model, assuming independence of factors, is deemed unrealistic in our case. Instead, we focus on identifying factors learned in the latent space, which distinguishes our approach from [1] and during the training phase of VAE. In addition, our contrastive objective is not similar to those of [1]. In our contrastive objective, there are two terms, one for invariance and the other for alignment of latent variables.
>
> Please if you have any further questions about our paper or its underlying assumptions, we will be happy to provide detailed answers.
>
> Best,
>
> Authors
>
> [1] Ren and al. "Learning Disentangled Representation by Exploiting Pretrained Generative Models: A Contrastive Learning View", ICLR (https://openreview.net/forum?id=j-63FSNcO5a)
>
> [2] https://openreview.net/forum?id=oUeYSTIhpE
>
> [3] https://openreview.net/forum?id=GAXedKmbFZ

---

> > ### Author Response · Authors · 2023-11-23
> > **Thank you for your review! Updates to our rebuttal**
> >
> > Dear Reviewer a2rd,
> >
> > Hope you're well. We'd like to revisit earlier points and seek your insights. Kindly take a moment to review our rebuttal, noting added experiments and clearer comment explanations. With sincere appreciation for your comprehensive evaluation, we kindly ask for a reconsideration of the score, taking into account the notable enhancements we've made.
> >
> > Thank you for your time.
> >
> > Best regards,
> >
> > Authors,

---

### Author Response · Authors · 2023-11-23

We express our gratitude to the reviewers for their valuable insights. We appreciate the positive acknowledgment of the problem we explored. In response to the constructive feedback, we've updated our manuscript, focusing on improving the formulation of our proposed methods for clarity. Detailed responses to individual reviewers, along with key changes:

- Enhanced our discussion on moving from the assumption of independent factorization to Independence-of-support factorization, including a comparison with the use of Hausdorff distance.

- Improved the presentation of the augmentation data section and the pipeline for sampling correlated appliances, addressing concerns from reviewer jyEy.

- Clarified our approach using the complete name, **D**isentangling based on **I**ndependence-**O**f-**S**upport via **C**ontrastive (**Diosc**), to avoid confusion with similarly named existing works but pursue distinct objectives.

- Revised the presentation of optimal hyperparameter choices in Appendix B and provided additional details on the number of positive samples used in the three considered scenarios (A,B,C) for six appliances.

We have thoroughly responded to all comments and trust that our revisions underscore the value of our work. Thank you for your time and consideration.

Best regards,

Authors

---

### Meta-Review · Area_Chair_31cp · 2023-12-13

**Metareview:**

This paper investigates the problem of disentangled representation learning from time series, specifically for home appliance electricity usage, which may help users understand and optimize their electricity consumption. The authors noticed that the disentangled processes might be dependent, and exploit weakly supervised contrastive learning to facilitate representation generalization across correlated scenarios and new households.

The paper nicely uses pairwise similarity in VAE to allow dependence in latent representations and l-variational inference layers with self-attention mechanism to address temporal dependences. The readability of the paper can be improved, according to the comments from two reviewers, and the experimental results should be made more concrete.

**Justification For Why Not Higher Score:**

The readability of the paper can be improved, according to the comments from two reviewers, and the experimental results should be made more concrete.

**Justification For Why Not Lower Score:**

The paper nicely uses pairwise similarity in VAE to allow dependence in latent representations and l-variational inference layers with self-attention mechanism to address temporal dependences.

---

### Decision · Program_Chairs · 2024-01-16

Accept (poster)